# Efficacy of Pixel-Level OOD Detection for Semantic Segmentation

## Abstract

The detection of out of distribution samples for image classification has been widely researched. Safety critical applications, such as autonomous driving, would benefit from the ability to *localise* the unusual objects causing the image to be out of distribution. This paper adapts state-of-the-art methods for detecting out of distribution images for image classification to the new task of detecting out of distribution pixels, which can localise the unusual objects. It further experimentally compares the adapted methods on two new datasets derived from existing semantic segmentation datasets using PSPNet and DeeplabV3+ architectures, as well as proposing a new metric for the task. The evaluation shows that the performance ranking of the compared methods does not transfer to the new task and every method performs significantly worse than their image-level counterparts.

## 1 Introduction

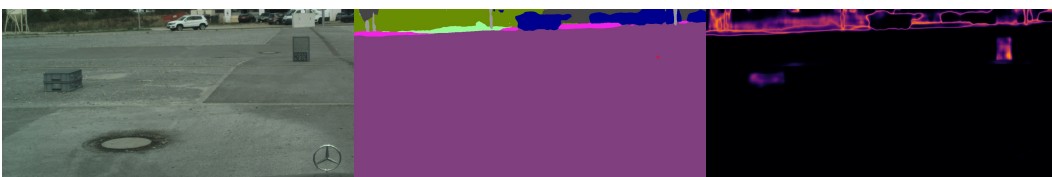

Figure 1: Image from the LostAndFound dataset (Pinggera et al., 2016), where two unlikely objects (storage crates) are almost entirely incorrectly predicted to be road. The Max Softmax method clearly highlights these crates as OOD. (best viewed in colour)

Many applications using machine learning (ML) may benefit from out of distribution (OOD) detection to improve safety. When inputs are determined to be out of distribution, the output of an ML algorithm should not be trusted. A large body of research exists for detecting entire images as OOD for the task of image classification. Image-level OOD detection outputs a classification for the entire image; this coarse level of detection may be inadequate for many safety critical applications, including autonomous driving. Most of the pixels in an image taken from an onboard camera will be in distribution (ID), *i.e.* an image of a road scene with cars, people, and roadway—but an unusual object that was not part of the training set may cause only a small number of OOD pixels. Extending the framework to semantic segmentation networks will allow each pixel to have an "in" or "out of" distribution classification. Applied to autonomous driving, groups of pixels classified as OOD would be considered as unknown objects. Depending on the location of the unknown objects, a planner would then proceed with caution or hand over control to a safety driver. Another application is automatic tagging of images with OOD objects, which would then be sent for human labelling. Figure 1 shows a failure case where OOD detection is beneficial. The two crates are predicted as road. The right image of this figure shows the result of pixel-level OOD detection using one of the proposed methods, which clearly identifies the unusual objects.

This paper adapts existing state-of-the-art image-level OOD detection methods to the new task of pixel-level OOD classification and compares their performance on a new dataset designed for this task. In addition to adapting the methods, we address the question of whether the best-performing image-level methods maintain their performance when adapted to the new task. In order to answer this question, we also propose pixel-level OOD detection performance metrics, drawing both on

existing image-level OOD detection and semantic segmentation performance metrics. Further, we design two new datasets for pixel-level OOD detection with test images that contain both pixels that are in distribution and pixels that are out of distribution, evaluated with two different network architectures—PSPNet (Zhao et al., 2016) and DeeplabV3+ (Chen et al., 2018). Somewhat surprisingly, our evaluation shows that the best performing pixel-level OOD detection methods were derived from image-level OOD detection methods that were not necessarily the best performing on the image-level OOD detection task. In summary, the contributions of this paper are the following:

- adaptation of image-level OOD detection methods to pixel-level OOD detection and their evaluation;
- training and evaluation datasets for pixel-level OOD detection evaluation derived from existing segmentation datasets; and
- a new metric for pixel-level OOD detection, called MaxIoU.

## 2 ADAPTING OOD METHODS

We use two criteria to select existing image-level OOD detection methods to adapt to pixel-level OOD detection. First, the candidate methods are top performers on image classification datasets. Second, they must be computationally feasible for semantic segmentation. Max Softmax (Hendrycks & Gimpel, 2016), ODIN (Liang et al., 2017), Mahalanobis (Lee et al., 2018) and Confidence (DeVries & Taylor, 2018) fit both criteria. The Entropy (Mukhoti & Gal, 2018), Sum of Variances (VarSum) (Kendall et al., 2015), and Mutual Information (Mukhoti & Gal, 2018) methods do not meet the first criterion, but are included as an existing uncertainty baseline for the pixel-level OOD classification. An example method that does not meet the second criterion is an ensemble method by Vyas et al. (2018), with an ensemble of K leave-out classifiers. In general, images and architectures for semantic segmentation are larger than for image classification, and therefore an ensemble method is much less feasible for segmentation than classification due to GPU memory limitations. Generative Adversarial Networks (GANs) and Auto-Encoders (AEs) are often used in the literature as a method for OOD detection. GANs and AEs are excluded to limit the scope of this work.

Table 1 briefly describes the selected image-level OOD detection methods and any architecture modifications necessary to adapt them to OOD detection.[1] The Dim reduction modification is an additional penultimate layer that is a $1 \times 1$ convolution reducing the depth to 32. DeeplabV3+ has a much larger feature extractor than PSPNet, therefore due to hardware limitation, the Mahalanobis method is not evaluated on this architecture. Each original/adapted method produces a value that can be thresholded to predict whether an image/pixel is OOD. All metrics used in our evaluation are threshold independent, therefore no thresholds are discussed in this section. See Appendix A for more detailed description of each OOD detection method.

| Method Name | Modifications | Short Description |
|---|---|---|
| Max Softmax | $\times$ | the maximum predicted probability after applying the softmax function |
| ODIN | $\times$ | augments the Max Softmax by dividing by a temperature value before applying the softmax function. Higher temperature values make the predictions more uniform. |
| Mahalanobis | Dim reduction | models the activations of the penultimate layer over the training set and computes the Mahalanobis distance to the class mean for each spatial location. |
| Confidence | Extra branch | adds a second branch to the output that learns the confidence of each pixel via an extra loss during training. |
| Entropy | $\times$ | computes the Shannon entropy of the output after the softmax function |
| VarSum | Dropout layers | computes the sum of variances of the predictions |
| Mutual Information | Dropout layers | the entropy of the mean prediction minus mean entropy of all predictions over multiple runs |

Table 1: List of methods compared, modifications required, and a short description.

ODIN has a temperature value hyperparameter that is optimised over a held-out training sample of OOD images. This should be considered when comparing to other methods that do not have access

---

[1]Code for each method is attached to the submission.

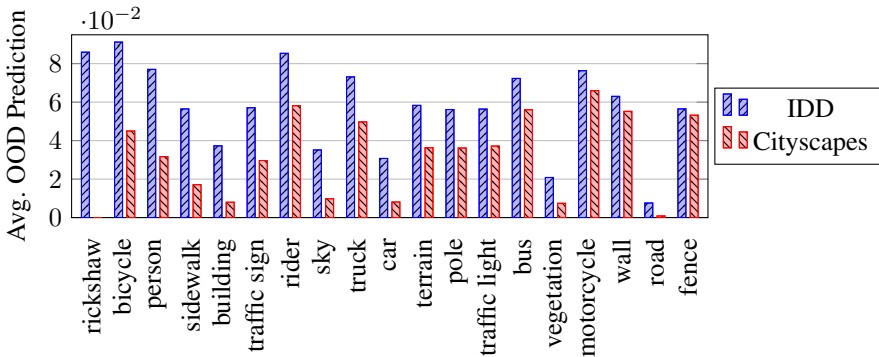

Figure 2: Average OOD prediction value of the Entropy method across all images and pixels for each class. The classes are sorted by difference between IDD and Cityscapes from most to least, left to right. The train class is removed as the IDD dataset doesn't have any instances.

to OOD data prior to evaluation. This unfair comparison has been criticised in the past (Hendrycks et al., 2018). Most methods have a input perturbation step; however, this is removed as it also requires access to OOD data before evaluation.

## 3    ADAPTING DATASETS

Previous work on image-level OOD detection designates a dataset used for training as the ID dataset, and the dataset for testing as the OOD dataset. This framework is not easily extended to semantic segmentation, as any two datasets may share features with the ID dataset. For example, people may exist in both datasets, but one is indoor scenes and the other is outdoor scenes. Blum et al. (2019) attempt to get around this issue by inserting animals from the COCO dataset (Lin et al., 2014) or internet images into the Cityscapes dataset. The lack of realism of the inserted images (improper scale and hard boundaries) make the created dataset insufficient for OOD detection. The remainder of this section describes the datasets used in this work and any modifications required.

### 3.1    TRAINING

The dataset used for training the weights of all networks is the unmodified Cityscapes dataset (Cordts et al., 2016). It is the most prominent dataset for training and benchmarking semantic segmentation for road scene images.

### 3.2    EVALUATION

The diverse SUN dataset (Xiao et al., 2010) and the India Driving Dataset (IDD) (Varma et al., 2018) are used as the main evaluation datasets. The main motivation for using the SUN dataset is that it has a large variety of scenes (*e.g.* street, forest, and conference room), as well as a large variety of labels (*e.g.* road, door, and vase). In total there are 908 scene categories and 3819 label categories. Anonymous label submissions are ignored, as their validity is not confirmed. However, the SUN dataset is not very realistic in terms of what a vehicle camera would see. The IDD dataset is an autonomous vehicle dataset that has all the same labels of Cityscapes with the addition of the auto-rickshaw class. Although the classes are the same, the instances tend to be OOD, due to different building architecture, road infrastructure, *etc*. This shift in features leads to higher average values OOD predictions as shown in Figure 2. Therefore only the car class as ID and auto-rickshaw class as OOD are used for evaluation.

The SUN dataset labels have to be modified before the dataset can be used for evaluating OOD detection.[2] The approach for modifying this dataset is similar to the open world dataset created by Pham et al. (2018). Let $C = \{\text{person}, \text{car}, \text{ignore}, ...\}$ be the set of labels available in Cityscapes. Let $S = \{\text{person}, \text{roof}, \text{chair}, ...\}$ be the set of labels available in the SUN dataset. Ambiguous classes

---

[2]Code for converting the SUN dataset is will be released publicly upon acceptance.

such as "wall" (can appear indoor and outdoor) or "path" (a sidewalk or a dirt path in the woods) are sorted into a third set $A \subset S$. The map $\mathcal{M} : S \to C \cup \{OOD\}$ is defined as:

$$\mathcal{M}(s) = \begin{cases} s & \text{if } s \in C \setminus A \\ \text{ignore} & \text{if } s \in A \\ OOD & \text{otherwise} \end{cases} \tag{1}$$

For every image in the SUN dataset, each pixel label $l_i$ gets a new label $l_i' = \mathcal{M}(l_i)$. Pixels with the "ignore" class are given a weight of 0 during evaluation.

All the images in the SUN dataset have various dimensions. To prevent artefacts in the input after resizing, only images that have more than $640 \cdot 640 = 409,600$ pixels are used. All images are resized to the same size as Cityscapes ($1024 \times 2048$).

The SUN dataset is split into a train set and an evaluation set (25%/75% split). It is stressed that the training set is used only to select the best hyperparameters of the ODIN method.

Following previous work on image-level OOD detection, a synthetic dataset of random normally distributed images are used as well. The random normal noise is usually very easily detected by all methods, therefore Perlin noise (Perlin, 1985) images are used as well. Perlin noise is a smooth form of noise that has large "blobs" of colour that is much harder to detect and filter. Each of these datasets have the entire image labelled as OOD.

All OOD datasets used are mixed with Cityscapes evaluation sets. The resulting ratio of ID to OOD pixels for IDD/Cityscapes and SUN/Cityscapes is about 7.5:1 and 3:1 respectively. Since ODIN requires ODD data for hyperperameter tuning, Cityscapes training set is mixed with a held-out training set of OOD data in the same manner as the evaluation sets.

## 4    Performance Metrics

There are five metrics used to evaluate the performance of each model. The first four listed below are the same metrics usually used by previous works on OOD detection (Liang et al., 2017; Hendrycks & Gimpel, 2016; Lee et al., 2018; DeVries & Taylor, 2018). Since the output of semantic segmentation is so much larger than image classification, the below metrics must be approximated. This is done by using 400 linearly spaced thresholds between 0 and 1 and tracking all true positives (*TP*), true negatives (*TN*), false positives (*FP*), and false negatives (*FN*) for each threshold. Each pixel contributes to one of *TP, TN, FP, FN* for a given threshold, based on the ground truth and OOD prediction value – accumulated across all images.

- **AUROC** – Area under the receiver operating characteristic (ROC) curve. The ROC curve shows the false positive rate (FPR) $\frac{FP}{FP+TN}$ against the true positive rate (TPR) $\frac{TP}{TP+FN}$. The area under this curve is the AUROC metric.
- **AUPRC** – Area under the precision recall (PR) curve. The PR curve shows the precision $\frac{TP}{TP+FP}$ against the TPR (or recall). The area under this curve is the AUPRC.
- **FPRatTPR** – FPR at 95% TPR. Extracted from the ROC curve, this metric is the FPR value when the TPR is 95%.
- **MaxIoU** – Max intersection over union (IoU). IoU is calculated by $\frac{TP}{TP+FP+FN}$. MaxIoU is the maximum IoU over all thresholds.

A common metric in the literature, coined *detection error* (Liang et al., 2017), is excluded as it is a linear transformation of the FPRatTPR metric. Therefore, it adds no useful information.

### 4.1    MaxIoU

The inspiration for MaxIoU issued from the semantic segmentation community. Mean intersection over union (mIoU) is the canonical performance metric for semantic segmentation. MaxIoU is similar, but targets tasks that are threshold dependant. Thresholds selected by MaxIoU punish false positives more than AUROC. The optimal threshold is generally greater, resulting in fewer positive predictions than for AUROC. To verify that the MaxIoU is complimentary to AUROC, the optimal

thresholds selected by each were experimentally compared. The mean absolute difference between the threshold selected via Youden index (Youden, 1950) and that chosen for MaxIoU was found to be 0.039.

## 5 NETWORK ARCHITECTURE

PSPNet (Zhao et al., 2016) and DeeplabV3+ (Chen et al., 2018) network architectures are used with the Resnet (He et al., 2015) and Xception (Chollet, 2017) feature extractors respectively. The two driving factors for these networks are: near top performance on the Cityscapes benchmark (Cordts et al., 2016) (2.6 and 1.7 less mIoU than the top scoring algorithm) and the final operation before the softmax classification layer being a bi-linear upsample. The upsample ensures that any method's OOD prediction will be directly correlated with the pixel prediction from the softmax layer, as most methods rely on the softmax layer. There is a clear relationship between any spatial location in the penultimate layer and an output pixel.

The Xception feature extractor is much larger than the Resnet – therefore due to hardware limitations, the space intensive Mahalanobis method is only evaluated on PSPNet.

## 6 EXPERIMENTS

There are two main research questions that this paper focuses on. Each question has an associated experiment.

- **RQ1:** *Do the required modifications to architecture and loss functions negatively affect the semantic segmentation performance of the network?*

- **RQ2:** *Which OOD detection method performs the best?*

To answer **RQ1**, we evaluate the semantic segmentation performance on Cityscapes using the standard class mean intersection over union (mIoU) metric. As long as the performance drop of modified networks is not too large, the modifications do not interfere with the original task. **RQ2** is answered by evaluating each method on the datasets described in Section 3.

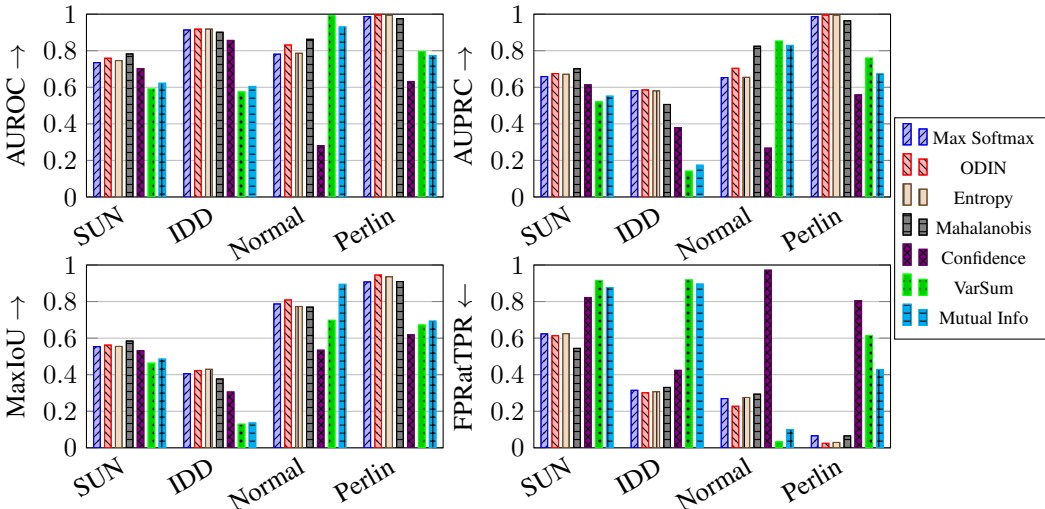

Figure 3: Comparison of methods on different datasets using the **PSPNet** architecture. The arrow on the y-axis label indicates if a larger value is better (↑) or a smaller value is better (↓). Each group of bars are labelled by the dataset used for evaluation.

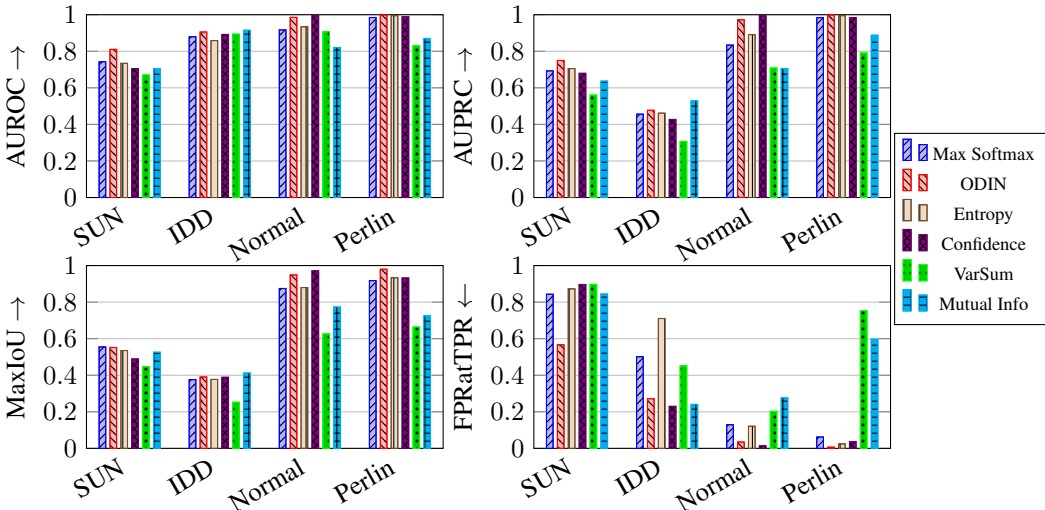

Figure 4: Comparison of methods on different datasets using the **DeeplabV3+** architecture. The arrow on the y-axis label indicates if a larger value is better (↑) or a smaller value is better (↓). Each group of bars are labelled by the dataset used for evaluation.

# 7 RESULTS AND DISCUSSION

## 7.1 ARCHITECTURE MODIFICATIONS (RQ1)

Table 2 shows the mIoU of PSPNet and DeeplabV3+ trained on Cityscapes and evaluated on Cityscapes. The modifications degrade performance only slightly. The maximum performance degradation between the unmodified network and any modified version is small, at $2.4\%$ for PSP-Net and $8.6\%$ for DeeplabV3+. As stated before, Mahalanobis with DeeplabV3+ is too memory intensive, therefore it is not included.

| Network | mIoU | Network | mIoU |
|---|---|---|---|
| PSPNet | **0.6721** | DeeplabV3+ | **0.7933** |
| PSPNet + dim reduce | 0.6701 | DeeplabV3+ + dim reduce | - |
| PSPNet + dropout | 0.6593 | DeeplabV3+ + dropout | 0.7252 |
| PSPNet + confidence | 0.6560 | DeeplabV3+ + confidence | 0.7656 |

Table 2: mIoU values for modified PSPNet and DeeplabV3+ architectures showing efficacy for original segmentation task. Network modifications are described in Table 1.

## 7.2 METHOD PERFORMANCE (RQ2)

Figure 3 shows the comparison of the performance of the different methods using the PSPNet architecture. Each graph shows a different metric (*c.f*. Section 4) for each dataset. Max Softmax, ODIN, and Mahalanobis follow the same trend as their image classification counterparts with the SUN dataset, increasing in performance in that order. However, for the IDD dataset the order is Mahalanobis, Max Softmax, then ODIN, in increasing performance. For image-level OOD detection, the Confidence method outperforms the Max Softmax baseline. However, for pixel-level OOD detection it is worse. For real datasets, VarSum is worse than Mutual Information, but better on the synthetic datasets.

Across each metric, VarSum has the biggest performance increase from the modified SUN and IDD datasets to the random normal dataset, moving from worst to near top performance. This however is not the same for the Perlin noise dataset, as the low frequency noise is not easily filtered. The Confidence method seems to mostly learn to highlight class boundaries, as that is where prediction errors are likely to occur. Therefore the prediction loss force lower confidence levels. This makes it less suitable for the random normal and Perlin noise datasets, where the network predicts one single class for the majority of the input.

Figure 4 shows the comparison of the performance of the different methods using the DeeplabV3+ architecture. With respect to AUROC, AUPRC, and MaxIoU on the IDD dataset Mutual Information

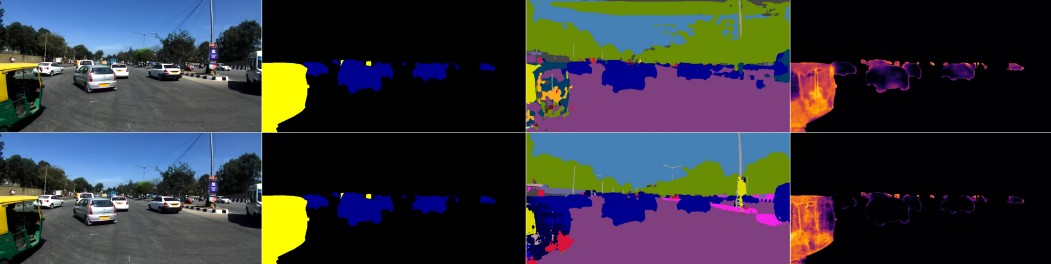

Figure 5: Comparison of the Mutual Information method on an IDD dataset image, successfully predicted. The top row is using the PSPNet architecture, the bottom row is using the DeeplabV3+ architecture. The columns from left to right are: input image, ground truth, class prediction (Cityscapes colours), ODD prediction. The OOD prediction is masked to only cars and auto-rickshaws. Best viewed in colour.

has the best performance, followed by ODIN. On all datasets and metrics, VarSum has the worst or is near worst performance. Confidence has much better performance with the DeeplabV3+ architecture than the PSPNet architecture, especially on the random datasets. The results for DeeplabV3+ are much less conclusive than PSPNet as the relative ordering across the different metrics changes significantly.

The metrics values reported in the original works for the image-level OOD detection for all methods with image-level OOD detection counterparts are very close to 1 (~0.95 and above). However, in our experiments, the results are much less than 1. One peculiar result is the significant difference in performance between the Confidence method on DeeplabV3+ and PSPNet on the random normal and Perlin noise datasets.

ODIN performs well on all datasets and architectures. The performance of this method is aided by its access to OOD data during hyperparameter tuning, however.

## 7.3 QUALITATIVE DISCUSSION (RQ2)

Figure 5 shows a comparison of the Mutual Information method using both the PSPNet and DeeplabV3+ architecture. Mutual information is one of the worst performing on PSPNet and the best performing on DeeplabV3+. There are two major observations to note. The first is that both networks label different parts of the auto-rickshaw as different classes. The second is the increase in OOD prediction of all cars from DeeplabV3+ to PSPNet, while the OOD prediction of the auto-rickshaw remains approximately the same. Both architectures successfully predict the majority of the pixels correctly; however, the separation is better for DeeplabV3+.

Figure 6 shows a comparison of the Entropy method using both the PSPNet and DeeplabV3+ architecture. The auto-rickshaw is mistaken by both architectures as a car, therefore the resulting OOD prediction is similar to other cars in the scene. This failure highlights a more general failure of pixel-level OOD detection methods. When the network is confident about a wrong prediction, the OOD methods are fooled as well.

The drop in performance for pixel-level OOD detection is likely due to features that cause large disruptions at the pixel-level, but would not affect an entire image; for example, shadows, occlusion, and far away objects. Figure 7 shows an example of shadows and far away objects in the bottom row. At the end of the road, most pixels are high OOD values as well as the right side of the scene, which is in the shade of a building. The top row of Figure 7 shows an interesting failure case of a flooded road being predicted as road with a low OOD value.

As can be seen in all example outputs, class boundaries are highlighted. A classical computer vision algorithm was developed, using a series of erosion, dilation and other filters to remove these boundaries. In general performance was increased; however, the increase was on the order of $10^{-3}$.

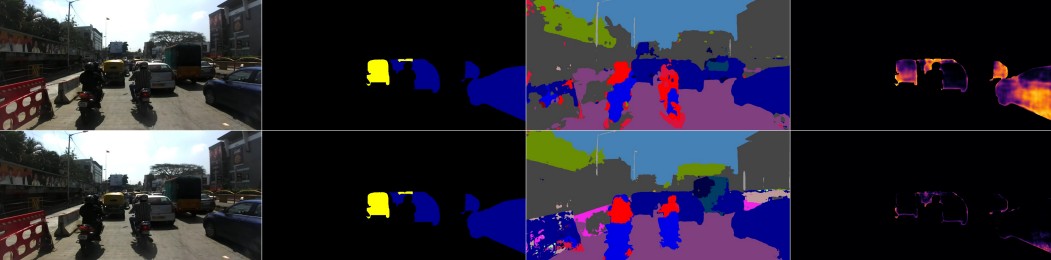

Figure 6: Comparison of the Entropy method on an IDD dataset image, successfully predicted. The top row is using the PSPNet architecture, the bottom row is using the DeeplabV3+ architecture. The columns from left to right are: input image, ground truth, class prediction (Cityscapes colours), ODD prediction. The OOD prediction is masked to only cars and auto-rickshaws. Best viewed in colour.

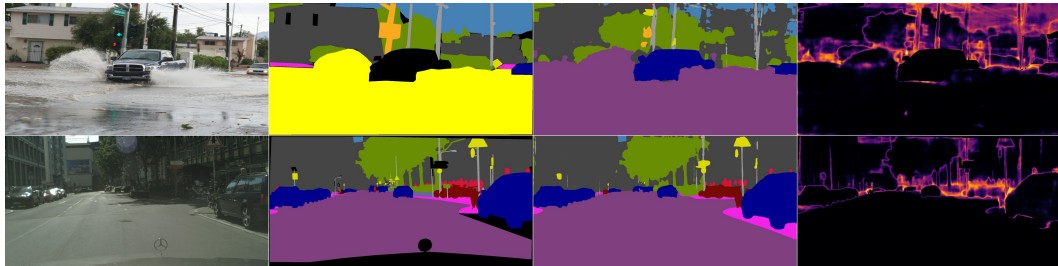

Figure 7: Examples from the SUN (top row) and Cityscapes (bottom row) datasets using PSPNet. The columns from left to right are: input image, ground truth, class prediction (Cityscapes colours), Entropy ODD prediction.

## 8 RELATED WORK

To our knowledge, there are very few works researching pixel-level OOD detection. Blum et al. (2019) create a dataset by overlaying animals and objects from the COCO dataset (Lin et al., 2014) on top of the Cityscapes dataset (Cordts et al., 2016), this however lacks realism. This new dataset is tested with various OOD detection methods. Bevandic et al. (2018) train a segmentation network with two datasets—one ID and one OOD dataset. The network learns to classify between the two on a per pixel-level. This is compared to the Max Softmax baseline. The major flaw in this method is that the network learns its weights from OOD samples.

There are some commonalities to Active Learning for semantic segmentation (Gorriz et al., 2017; Mackowiak et al., 2018). These studies attempt to use some form of output uncertainty to choose which images are best for training/labelling in order to reduce the number of training examples needed. They produce a heat map similar to the OOD detection output. These heat maps are then aggregated across a whole image to produce a score for the entire image.

Pham et al. (2018) create an open world dataset. Known object labels are drawn from the COCO dataset (Lin et al., 2014), and labels drawn from the NYU dataset (Silberman et al., 2012) are relabelled as unknown if the class doesn't exist in COCO. This methodology is very similar to the modified SUN dataset in Section 3. A generic object instance level segmentation algorithm is developed, based on a class specific object detector, a boundary detector and simulated annealing, and is evaluated on the new dataset. This approach splits the image into visually distinct connected regions, but is too slow for real-time applications.

## 9 CONCLUSIONS AND FUTURE WORK

Several methods for detecting OOD pixels were adapted from image-level OOD detection, as well as a pixel uncertainty estimation. These methods were compared using metrics previously established by OOD detection works, as well as a new metric that has roots in the semantic segmentation task. This paper also contributed two new datasets for pixel-level OOD classification derived from semantic segmentation datasets that have common classes but also unique ones.

There is great room for improvement for pixel-level OOD detection. One shortcoming for all the methods compared in this paper is the ability to distinguish between class boundary pixels and OOD pixels. We tested classical computer vision techniques that could be used to visually fix this problem, but the performance increase was negligible. The ODIN and Mahalanobis methods have the best performance with PSPNet and SUN dataset, beating the VarSum, Mutual Information, and Confidence methods by a significant margin. However, Mutual Information has the best performance with DeeplabV3+ and the IDD dataset, with the other methods following closely. Therefore the ODIN, Mahalanobis, and Mutual Information methods should be considered the baseline for further research in pixel-level OOD detection.

Understanding the faults of pixel-level OOD detectors is crucial for progress. This would include categorising the failure cases of a detector. For example, understanding why a flooded road is not highlighted, and what makes that different to shadows falsely being highlighted.

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

## A  OOD Detection Methods

Each subsection uses notation close to the original works, so that readers can refer to said works for more detail. The appendix also provides additional detail. Notation should not be carried between sections unless explicitly stated. Assume the neural network is a function $f$, and input image is $x$. The set of pixels $P$ will be used with typical subscripts $i, j \in P$. For example $f(x)_i$ is the $i^{\text{th}}$ pixel of the result of $f(x)$.

Note that most methods have a input perturbation step. They are included here for completeness, however, they are not used in any evaluation as it requires access to OOD data. This access to OOD data has been criticised as an unfair comparison to methods that do not require access to OOD data.

### A.1 VARSUM

At test time, dropout (Srivastava et al., 2014) can be used as an estimate for model uncertainty. A multiplier of 0 or 1 is randomly chosen for each neuron at test time. (Kendall et al., 2015) use the variation in predictions of a model that uses dropout test time to compute an estimate of model uncertainty. The output is a variance per class, the sum of these variances is the estimate of model uncertainty. This is performed per pixel to get the value $s_i$ for pixel $i \in P$.

### A.2 MAX SOFTMAX

Hendrycks & Gimpel (2016) show that the max softmax value can be used to detect OOD examples as a baseline for image classification. The softmax function is defined as:

$$S_j(x) = \frac{e^{x_j}}{\sum_k e^{x_k}} \tag{2}$$

$$S_{\hat{y}}(x) = \max_j S_j(x) \tag{3}$$

Where $S_{\hat{y}}(x)$ is the max softmax value and $\hat{y}$ is used to signify the chosen index.

Given a prediction $p_i = f(x)_i$, the max softmax value for each pixel $i$ is $v_i = S_{\hat{y}_i}(p_i)$, and that value is used to determine if that pixel is OOD.

### A.3 ODIN

Liang et al. (2017) create a similar method to the max softmax value, dubbed ODIN. This method adds temperature scaling and input preprocessing. The softmax function in Equation 3 is modified to include a temperature value $T$:

$$S(x; T) = S\left(\frac{x}{T}\right) \tag{4}$$

$$S_{\hat{y}}(x; T) = \max_j S_j(x; T) \tag{5}$$

Liang *et al.* found that perturbing the input in the direction of the gradient influences ID samples more than OOD samples, thus separating ID and OOD examples more. The input preprocessing step is:

$$\tilde{x} = x - \epsilon \cdot \text{sign}\left(-\sum_{i \in P} \nabla_x \log S_{\hat{y}}(f(x)_i; T)\right) \tag{6}$$

where $\epsilon$ is a hyperparameter chosen from a set of 21 evenly spaced values starting at $0$ and ending at $0.004$. The best temperature value is chosen from a predefined set of temperatures $\{1, 2, 5, 10, 20, 50, 100, 200, 500, 1000\}$.

The temperature-scaled and preprocessed max softmax score $v_i = S_{\hat{y}}(\tilde{x}; T)_i$ for pixel $i$ is used to predict if that pixel is OOD.

### A.4 MAHALANOBIS

Lee et al. (2018) use the Mahalanobis distance for detecting OOD samples. The Mahalanobis distance is the number of standard deviations a vector is away from the mean, generalised to many dimensions.

Each feature vector at a spatial location in the penultimate layer is assumed to be normally distributed. Each distribution is parameterised by the pixel class mean $\mu_{ci}$ and global class covariance $\Sigma_c$ for pixel $i$ and class $c$. The global class covariance is computed for all pixels of a given class, independent of their location. Initial tests showed that using pixel class means and a global class covariance has better performance than global or pixel class mean and pixel class covariance, therefore they are used throughout. The labels are resized to match the height and width of the penultimate

layer using nearest neighbour interpolation. The two quantities $\mu_{ci}$ and $\Sigma_c$ are computed as follows:

$$\mu_{ci} = \frac{1}{N_c} \sum_{j:y_j=c} f_l(X_j)_i \tag{7}$$

$$\Sigma_c = \frac{1}{N_c} \sum_{i,j:y_j=c} (f_l(X_j)_i - \mu_{ci})(f_l(X_j)_i - \mu_{ci})^T \tag{8}$$

where $N_c$ is the number of examples of class $c$, and $X_j$ is the $j^{\text{th}}$ example of the training dataset $X$. $f_l$ is the output of the $l^{\text{th}}$ layer of the network $f$. Here $l$ is the second to last layer.

Each spatial location has a class distance and minimum distance computed by:

$$M_c(x)_i = \sqrt{(f_l(x)_i - \mu_{ci})^T \Sigma_c^{-1} (f_l(x)_i - \mu_{ci})} \tag{9}$$

$$M(x)_i = \min_c M_c(x)_i \tag{10}$$

This increases the number of matrix multiplications required to compute each pixel distance. Due to hardware memory limitations, a dimensionality reduction layer is needed after the penultimate layer, reducing the depth from 512 to 32 via a $1 \times 1$ convolution. An input preprocessing step is also performed. The new input $\tilde{x}$ is computed by:

$$\tilde{x} = x - \epsilon \cdot \text{sign} \left( - \sum_{i \in P} \nabla_x M(x)_i \right) \tag{11}$$

Instead of a logistic regression layer, the minimum distance is normalised to have zero mean and unit variance. The sigmoid function $\sigma$ is applied to clamp to the interval $[0, 1]$:

$$v_i = \sigma \left( \frac{M(x)_i - \mu}{s} \right) \tag{12}$$

where $\mu$ and $s$ are the mean and standard deviation of all $M(x)$ computed over the whole training dataset. $v_i$ is used to determine if pixel $i$ is OOD. The prediction values are resized, with bi-linear interpolation, to the original input size.

## A.5 CONFIDENCE ESTIMATION

DeVries & Taylor (2018) train a secondary branch of an image classification network to output a single confidence value. This confidence value is learned via a regularization loss. The loss gives "hints" to the network when the confidence is low, and it penalizes low confidence.

Similar to the image classification method, a secondary branch is added to the network $f$. Therefore $c_i, p_i = f(x)_i$, where $c_i$ is the confidence value and $p_i$ is the prediction vector for pixel $i \in P$. The new branch is trained by creating a new prediction $p_i'$ as:

$$b_i \sim B(0.5) \tag{13}$$

$$c_i' = c_i \cdot b_i + (1 - b_i) \tag{14}$$

$$p_i' = c_i' \cdot p_i + (1 - c_i') \cdot y_i \tag{15}$$

where $B$ is a Bernoulli distribution and $y_i$ is the one hot ground truth vector for pixel $i$. The mean negative log likelihood is then applied to the new $p_i'$ as well as a regularization term is added to force each $c_i$ to 1 (*i.e.* high confidence):

$$\mathcal{L}_t = \frac{1}{|P|} \sum_{i \in P} - \log(p_i') y_i \tag{16}$$

$$\mathcal{L}_c = \frac{1}{|P|} \sum_{i \in P} - \log(c_i) \tag{17}$$

$$\mathcal{L} = L_t + \lambda L_c \tag{18}$$

where $\mathcal{L}$ is the total loss that is used to train the network, and $\lambda$ is a hyperparameter. $\lambda$ is $0.5$ in all experiments.

At test time a preprocessing step is applied to each pixel, and is computed using the gradients of the $L_c$ loss. Note that the adapted $L_c$ sums over all the pixel confidence predictions $c_i$, therefore the gradient implicitly sums over output pixels as well.

$$\tilde{x} = x - \epsilon \cdot \text{sign}\left(\nabla_x L_c\right) \tag{19}$$

Let $\tilde{p}_i, \tilde{c}_i = f_i(\tilde{x})$, then $v_i = \tilde{c}_i$ is used to determine if a pixel is OOD.

### A.6 ENTROPY

Shannon entropy (Shannon, 1948) is an information theoretic concept, that is used to determine how much information a source contains. The entropy equation is $H : \mathbb{R}^n \to \mathbb{R}$:

$$H(x) = -\sum_i x_i \cdot \log x_i \tag{20}$$

Since $H$ was developed for probabilities, $x$ must behave like a probability distribution meaning:

$$\sum_i x_i = 1 \tag{21}$$

$$\forall i, x_i >= 0 \tag{22}$$

Hendrycks et al. (2018) train an image-level classifier with a third outlier dataset that is disjoint from both the training and OOD dataset. An auxiliary loss is added minimising the entropy of predictions of outlier images. The network learns to predict uniform values for all classes. The max softmax value is then used at test time to determine if a sample is OOD.

The entropy function is applied to the softmax output, which satisfies the properties in Equations 21 and 22. Let $P = f(x)$ then

$$S = \text{softmax}(P) \tag{23}$$
$$v = H(S) \tag{24}$$

$v_i$ is used to determine if pixel $i$ is OOD.

### A.7 MUTUAL INFORMATION

Mukhoti & Gal (2018) develop Bayesian uncertainty estimation methods and evaluation metrics to evaluate if higher uncertainty is correlated with higher error rates. The estimation can be used for OOD detection as well, similar to the Entropy and VarSum methods. There are two quantities required. Predictive entropy:

$$\mathbb{H}(y|x) = -\sum_k p(y = k|x) \cdot \log[p(y = k|x)] \tag{25}$$

and Aleatoric Entropy:

$$\mathbb{AE}(y|x) = -\frac{1}{T} \sum_{k,i} p(y = k|x, \omega_i) \cdot \log[p(y = k|x, \omega_i)] \tag{26}$$

In both equations, $x$ is the input and $y$ is the prediction. $\omega_i$ is the set of weights selected by dropout at iteration $i$. Mutual information is then

$$\mathbb{MI}(y|x) = \mathbb{H}(y|x) - \mathbb{AE}(y|x) \tag{27}$$

$\mathbb{MI}(y|x)$ is used to determine if a pixel is OOD or ID.

