# OpenReview forum: "Efficacy of Pixel-Level OOD Detection for Semantic Segmentation"
_ICLR.cc/2020/Conference — Reject_

### Official Review · AnonReviewer2 · 2019-10-22
**Official Blind Review #2**

**Rating:** 3

**Review:**

The paper evaluates a variety of existing pixel-wise out-of-distribution detection methods in the task of semantic segmentation of road scenes. To do so, the paper introduces an evaluation protocol and applies it to two datasets (SUN and IDD) and two models (PSPNet and DeepLabV3+).

Strengths:
- The paper is well written with high quality visuals and plots
- The paper studies an important problem

Weaknesses:
- The contribution seems to be rather incremental (evaluating existing methods on 2 dataset) and some related work might be missing
- Although the analysis is well executed, it is not clear what the community learns from the paper


Although I enjoyed reading the paper, I'd lean towards rejection of the paper. My main concern are as follows:

It is not clear what the community learns after reading this paper. Is the difference between different approaches significant? Are the class boundary pixels the biggest problem? The paper would benefit strongly from providing some indications of future steps, e. g. how to improve OOD in semantic segmentation.

I'm missing a discussion between OOD and  and the prior work on uncertainty estimation in semantic segmentation (e. g. https://arxiv.org/pdf/1703.04977.pdf and the follow up works, or https://arxiv.org/pdf/1807.00502.pdf). It seems that uncertainty estimates for semantic segmentation could be applied to the tested scenarios off-the-shelf without the need for additional modifications. Is there any reason the paper did not include approaches for uncertainty estimation in semantic segmentation? In general, it would be useful to connect the tested OOD scenarios to other topics already studied in semantic segmentation such as: uncertainty estimation, outlier detection, and distribution shift in semantic segmentation. A paragraph drawing connections and highlighting the differences would make the paper stronger.

Other comments:

Section 3.2, "Therefore only the car class as ID..." I'm not sure I understand why car class is the only one considered ID for IDD. From Fig. 2, it seems that other classes such as bus, traffic light, pole, terrain, etc. could be also considered. Could the authors comment on this?

"The random normal noise is usually very easily detected by all methods, therefore Perlin noise images are used". However, when looking at the results Fig 3 and Fig 4 it feels like Normal random noise is harder than Perlin noise. Could the authors comment on this?

"All OOD datasets used are mixed with Cityscapes evaluation sets". Why it is important to add Cityscapes images to evaluation set? Wouldn't it be enough to use OOD datasets?

One suggestion of a plot that could jointly display the information from RQ1 and RQ2 would be to plot both of them in a one scatter plot (e. g. with ID IoU on x-axis and AUROC on the other).

Figures 3 and 4 show results for 6 methods, while Table 2 only displays 3 scenarios for 2 models. Table 3 would benefit from including all 6 models and using the same labels as in Figures. Moreover, it would be interesting expand Table 2 by including the performance of the segmentation on in distribution classes from IID and SUN datasets in addition to Cityscapes results.

**Experience Assessment:**

I have read many papers in this area.

**Review Assessment: Checking Correctness Of Derivations And Theory:**

N/A

**Review Assessment: Checking Correctness Of Experiments:**

I assessed the sensibility of the experiments.

**Review Assessment: Thoroughness In Paper Reading:**

I read the paper at least twice and used my best judgement in assessing the paper.

---

### Official Review · AnonReviewer1 · 2019-10-23
**Official Blind Review #1**

**Rating:** 1

**Review:**

- This paper proposes a comparative study of out-of-distribution (OOD) methods for semantic segmentation. To this end, authors extend networks designed for OOD image detection to accommodate for the segmentation task. For evaluation purposes authors create a new dataset and use 3 well-known metrics, as well as a new proposed metric.
- The paper is in general a bit dense to read and understand, since authors fail to explain many important details. Furthermore, the structure of the paper can be improved (e.g., related work is added at the end of the manuscript).
- Technical contribution of this work is insufficient. Authors merely employ ODD classification networks to address the pixel-level OOD task. Nevertheless, the motivation of those changes are never detailed. Furthermore, some other important aspects are not explained. For example, typically classification networks are adapted for segmentation adding a decoding path, so that the output result is a map of the same size of the input times number of classes. However, how the probability map for the pixel-level OOD predictions is never explained.
- Authors also mention that GANs and AEs are excluded to limit the scope of the paper. First, this reason is not convincing. Second, I believe that including all the significant works for the task-at-hand is more relevant. How these methods would work compared to the proposed networks?
- Unless I miss something, the adapted OOD versions degrade the semantic segmentation performance of original networks. Why not to use the original versions instead?
- Results are very unclear. Which dataset represents the OOD evaluation? Further, authors talk about results at image-level and pixel-level. Nevertheless, this is not detailed in the experimental section. Fig. 3 reports results of the different models at pixel-level. Where are the results of OOD at image-level? In addition, results are barely interpreted, and authors basically describe the values reported in the figures. I would appreciate a deeper interpretation of the results.
- I am curious to know why the confidence OOD approach has much better performance with DeeplabV3+ than with PSPNet. While PSPNet is among the worst performing models with confidence (sometimes the worst), Deeplab + confidence is typically top-ranked. Do the authors have any insight on this?
- There is no evidence that the proposed metric better models the OOD pixel-level performance than other standard metrics. Which are the results on this task achieved by mIOU instead?

**Experience Assessment:**

I have read many papers in this area.

**Review Assessment: Checking Correctness Of Derivations And Theory:**

I carefully checked the derivations and theory.

**Review Assessment: Checking Correctness Of Experiments:**

I carefully checked the experiments.

**Review Assessment: Thoroughness In Paper Reading:**

I read the paper thoroughly.

---

### Official Review · AnonReviewer3 · 2019-10-23
**Official Blind Review #3**

**Rating:** 3

**Review:**

# Summary

This paper puts forward a study over out-of-distribution (OOD) detection for semantic segmentation. OOD detection is an active area research which has recently dealt mostly with image level classification. For semantic segmentation the same conclusions might not apply since decisions must be taken for each pixel individually. To this effect the authors propose here to study this task over a set of architectures (PSPNet, DeepLabV3+), outlier datasets (SUN, Indian Driving Dataset, synthetic images) and multiple methods for OOD detection from recent works on image classification.
A major difficulty in OOD works is the definition of a relevant OOD dataset and evaluation setup, and the authors propose here a novel setup for this task by adjusting the SUN and IDD datasets as OOD for Cityscapes.
The experimental part is thorough with multiple evaluation metrics and some qualitative examples and discussion.

# Rating
Although the paper studies a meaningful and interesting problem, I would reject the paper for the following reasons (more detailed arguments some lines below):
1) I find that the proposed dataset setups for OOD detection are not adequate for semantic segmentation and thus alter the reliability of the results.
2) Other than the evaluation (which has some faults), there is no proposed method addressing this task and its challenges.
3) The proposed MaxIoU metric proposed is not discussed and compared in more detail against the usual ones for this task.


# Strong points
- This work is dealing with a highly interesting and challenging problem. Indeed there has been few studies in this area and defining proper techniques and evaluation setups is challenging.
- The authors consider a wide array of methods for evaluation and test them across two architectures and multiple real and synthetic datasets.

# Weak points
- There are two real datasets considered for OOD evaluation, but I consider there are some flaws in their utilization for OOD detection. First, the SUN dataset is quite different from Cityscapes with a significant domain gap (at least in the visual appearance and distributions of the classes) between the two datasets. Upsampling the SUN images to 5x their size in order to make them compatible to Cityscapes should increase the artefacts even further making it easier to spot the gap between SUN and Cityscapes. This means that there is a risk of having the OOD detector acting merely as domain classifier spotting whenever the domain is different from the one use for training.
This argument applies to IDD, although to a smaller extent as the datasets are both automotive, but again there are strong visual differences between samples of the same class across the two datasets.
The authors argue that they somehow take this argument into consideration (Figure 2) and select only the non-ID classes to perform evaluation. However in both networks the scene information is mixed into the representation (via pyramid pooling in PSPNet  respectively via a trous convolutions with different dilations in DeepLabV3+). So again, instead of an OOD detector we can end up doing domain classification.
It would be useful to see how is the classification performance changing for ID classes, e.g. how is a model trained on Cityscapes scoring for cars and other ID objects in IDD comparing to a model that is trained on IDD for the same classes. A big difference between these scores would correspond to a significant domain gap and in correlation with the OOD performance we might be able to take some more conclusions on the matter.
Ahmed and Courville[i] propose an interesting discussion on this type of problems and propose focusing on semantic anomaly detection, i.e. detecting different classes from the same dataset, to make sure the setting has practical interest. They propose a very basic technique to detect OOD in previous classification setups showing the limitations of previous OOD methods. I encourage the authors to check the arguments stated in that work.


- In my opinion, the Fishyscapes work from Blum et al. is unfairly dismissed here by considering only a part of the benchmark for which animals from COCO and internet are inserted over Cityscapes images. The authors argue that this lack of realism of the inserted images make this dataset insufficient for OOD detection. However in the first version of their paper, Blum et al. propose a mix of Foggy Driving, Foggy Zurich, WildDash and Mapillary as dataset for ODD detection, which is similar with the setup proposed here. Furthermore, the latest version of Fishyscapes includes the Lost & Found dataset (mentioned in Fig. 1 here) which is recorded in similar conditions with Cityscapes with the addition of a few small outlier objects used as OOD. This is a relevant dataset and work and I would adjust the critics brought to their work here. That paper has the same objectives and endeavors as the current submission.


- Although there are some discussions and experiments on multiple techniques there is not technical contribution mitigating all the limitations of previous OOD methods on classification and the challenges of OOD detection in semantic segmentation. This would have had greatly helped the paper.

- I find that the MaxIoU metric considered here is not sufficiently discussed and analysed to show its utility and the additional perspective it brings when evaluating along with the usual metrics.

## Other less important weak points
- The choice of dataset in Figure 1 , i.e. Lost & Found, can be misleading. This dataset is not further mentioned and evaluated in the rest of the paper. The image could be replaced with a qualitative results from the rest of the evaluation.

# Suggestions for improving the paper:
1) The current evaluation setting could have some flaws. I would propose some sanity checks and look at the classification performances over other ID classes, as suggested in the previous section

2) Evaluate on a setting similar to Fishyscapes Lost and Found, in which the dataset does not change much, but there are some novel objects.

3) Include a trivial OOD baseline in the spirit of [i] to show the utility of the proposed datasets for this task by being robust to such baselines.

4) Consider extending the breadth of OOD methods with Deep Prior Networks[ii] which have been shown to perform well on Fishyscapes for OOD detection.

5) Add a qualitative example with OOD detection on Perlin noise images

# References
[i] F. Ahmed and A. Courville, Detecting semantic anomalies, arxiv 2019 https://arxiv.org/abs/1908.04388

[ii] A. Malinin and M. Gales, Predictive uncertainty estimation via prior networks, NeurIPS 2018





**Experience Assessment:**

I have read many papers in this area.

**Review Assessment: Checking Correctness Of Derivations And Theory:**

I assessed the sensibility of the derivations and theory.

**Review Assessment: Checking Correctness Of Experiments:**

I carefully checked the experiments.

**Review Assessment: Thoroughness In Paper Reading:**

I read the paper thoroughly.

---

### Decision · Program_Chairs · 2019-12-19

**Decision:**

Reject

**Comment:**

This paper studies the problem of out-of-distribution (OOD) detection for semantic segmentation.

Reviewers and AC agree that the problem might be important and interesting, but the paper is not ready to publish in various aspects, e.g.,  incremental contribution and less-motivated/convincing experimental setups/results.

Hence, I recommend rejection.